# Further Exploration of an Upper Bound for Kemeny’s Constant

**DOI:** 10.3390/e27040384

**Published:** 2025-04-04

**Authors:** Robert E. Kooij, Johan L. A. Dubbeldam

**Affiliations:** 1Faculty of Electrical Engineering, Mathematics and Computer Science, Delft University of Technology, 2628 CD Delft, The Netherlands; j.l.a.dubbeldam@tudelft.nl; 2TNO (Unit ICT, Strategy & Policy, Netherlands Organisation for Applied Scientific Research), 2595 DA The Hague, The Netherlands

**Keywords:** Kemeny’s constant, effective graph resistance, random walks, spectral graph theory, pseudo-inverse Laplacian, 05C50, 05C75, 05C82

## Abstract

Even though Kemeny’s constant was first discovered in Markov chains and expressed by Kemeny in terms of mean first passage times on a graph, it can also be expressed using the pseudo-inverse of the Laplacian matrix representing the graph, which facilitates the calculation of a sharp upper bound of Kemeny’s constant. We show that for certain classes of graphs, a previously found bound is tight, which generalises previous results for bipartite and (generalised) windmill graphs. Moreover, we show numerically that for real-world networks, this bound can be used to find good numerical approximations for Kemeny’s constant. For certain graphs consisting of up to 100 K nodes, we find a speedup of a factor 30, depending on the accuracy of the approximation that can be achieved. For networks consisting of over 500 K nodes, the approximation can be used to estimate values for the Kemeny constant, where exact calculation is no longer feasible within reasonable computation time.

## 1. Introduction

Kemeny’s constant, a graph metric first proposed in 1960 [1], links random walks, Markov chains, and spectral graph theory; see, for instance, [2,3,4].

An intuitive way to understand Kemeny’s constant is by random walks on a graph, which was also how it was originally presented by Kemeny [1]. For an undirected connected graph with an adjacency matrix *A*, we can define a transition matrix $P_{ij}=A_{ij}/d_{i}$ for the transition from state *i* to *j*, where $d_{i}$ is the degree of node *i*. This defines an irreducible finite-state Markov chain in discrete time with an N×N transition matrix $P_{ij}$ [5]. If we also know the mean first-passage time matrix $m_{ij}$ denoting the average time to go from a vertex *i* to a vertex *j* (we take $m_{ii}=0$ by convention), the Kemeny constant is defined by(1)$$\begin{array}{cc} K(P) & =\sum_{j=1}^{N}\pi_{j}m_{ij}, \end{array}$$
where $\pi_{j}$ is the *j*-th component of the stationary solution of the random walk. The fact that K(P) does not depend on the index *i*, which can be interpreted as the starting state of the random walk and is therefore truly a constant, was discussed in a number of papers [6,7]. Hunter [8] and Kirkland [9] have analysed Relation (Equation 1) and established a connection with generalised matrix inverses.

The Kemeny constant also has an interpretation as a ‘mixing time’, which was originally proposed by Hunter in [7]. Here, we briefly repeat the demonstration that the Kemeny constant can be identified by a mixing time and show that this can be directly interpreted in terms of entropy. Let us define the ‘time to mixing’, *T*, of a Markov chain $\left\{X_{n}\right\}$ following [7], as the smallest index *k* at which $X_{k}=Y$, where *Y* is a random variable distributed according to the stationary distribution of the Markov chain $\left\{\pi_{j}\right\}$. We can now calculate the conditional expectation value of *T*, E[T|Y,X(0)=i],(2)$$\begin{array}{cc} E[T|Y,X(0)=i] & =\sum_{j}E\left[T,Y=j|X\left(0\right)=i\right]P\left[Y=j\right]\\ \; & =\sum_{j}E\left[T_{ij}|X\left(0\right)=i\right]\pi_{j}=\sum_{j}m_{ij}\pi_{j}=K\left(P\right), \end{array}$$
where $T_{ij}$ is the mean first-passage time for going from node *i* to node *j*.

Expression Equation (Equation 2) for the mixing time permits an interpretation in terms of *relative entropy* or Kullback–Leiber divergence D(p||π), which measures the distance between the distributions *p* and π; see also [10]. The relative entropy is defined as$$D_{n}\left(p||\pi \right)=\sum_{j}p_{j}\left(n\right)\log\frac{p_{j}\left(n\right)}{\pi_{j}}.$$Since $D_{n}\left(p||\pi \right)\geq 0$ with equality only when $p_{j}\left(n\right)=\pi_{j}$ for all j=1,⋯,N, the time to mixing can be interpreted as the smallest value of *n* for which the relative entropy $D_{n}\left(p||\pi \right)=0$.

Kemeny’s constant has recently also been suggested as a metric to identify bottleneck roads whose removal would greatly reduce the connectivity of the network [11] or as a metric to determine the ‘superspreader’ links that transmit disease between different communities [12].

It has already been established that there are several equivalent ways to express Kemeny’s constant: using effective graph resistance, random walks, spectral graph theory, and pseudo-inverse Laplacians; see [8].

The study of Kemeny’s constant is still an active and relevant research field, as was showcased by the mini-symposium “Kemeny’s constant on networks and its application”, which was organised as part of the 24th Conference of the International Linear Algebra Society, which took place in Galway, Ireland, 20–24 June 2022 [13] as well as recent papers addressing applications of Kemeny’s constant to different networks [14,15].

In 2017, Wang et al. [4] derived a closed-form formula for Kemeny’s constant, K(P) for a random walk on a graph *G* with *N* nodes and *L* edges, where the transition matrix was given by $P=\Delta^{-1}A\left(G\right)$, where A(G) is the adjacency matrix of *G* and Δ is a diagonal matrix containing the degrees of the nodes. In [4], it was shown that K(P) can be expressed in terms of the Moore–Penrose pseudo-inverse $Q^{†}$ of the Laplacian matrix of *G*, as (3)$$\begin{array}{cc} K(P) & =\zeta^{T}d-\frac{d^{T}Q^{†}d}{2L}, \end{array}$$
where the column vector $\zeta =\left(Q_{11}^{†},\;Q_{22}^{†},\;\ldots ,\;Q_{NN}^{†}\right)$ and $d\left(G\right)=\left(d_{1},\;d_{2},\;\ldots ,\;d_{N}\right)$ denotes the degree vector for the graph.

In [4], not only Equation (Equation 3) was derived, but also a closely connected upper bound:(4)$$K\left(P\right)\leq \zeta^{T}d-\frac{H(G)}{D\left(G\right)\mu_{1}\left(G\right)}\equiv K_{U}\left(P\right),$$
where D(G) is the average degree and $\mu_{1}\left(G\right)$ is the largest eigenvalue of the Laplacian matrix Δ(G)−A(G) corresponding to graph *G*. Here, Δ(G) denotes the diagonal matrix containing the degrees of the nodes. The heterogeneity index H(G), measuring the variability in the degrees of the nodes (see [16]) is defined as$$H\left(G\right)=\frac{1}{N}\sum_{i=1}^{N}{\left(d_{i}-D\left(G\right)\right)}^{2},$$
where $d_{i}$ is the degree of the i-th node.

It was shown in [17] that the upper bound given in Equation (Equation 4) is tight, meaning that we have an equality in Equation (Equation 4), for two classes of graphs, namely complete bipartite graphs and (generalised) windmill graphs. A windmill graph W(η,k) consists of η copies of the complete graph $K_{k}$, with each node connected to a common node. Two generalisations of windmill graphs were suggested by Kooij [18] in 2019. For both generalisations, we replace the central node, connecting all η copies of the complete graph $K_{k}$, with *l* central nodes. For the first generalisation, we assume that the *l* central nodes are all connected, i.e., they form a clique $K_{l}$. We call this a generalised windmill graph of Type I and denote it by $W^{\prime }\left(\eta ,k,l\right)$. For the second generalisation, we assume that the *l* central nodes have no connections among each other. We will refer to it as a Type II generalised windmill graph and denote it by $W^{\prime \prime }\left(\eta ,k,l\right)$. Figure 1 shows examples of a windmill graph and its two generalisations,

The aim of this paper is four-fold. First, we will consider a broad family of graphs, which contain complete bipartite and (generalised) windmill graphs as special cases, and show analytically that for these graphs, the bound Equation (Equation 4) is tight. Graphs in this family have in common that they are bimodal and have a diameter of two. However, we will also show that these conditions are not sufficient to ensure that Equation (Equation 4) is tight. Next, we compare the complexity of the computation of the upper-bound Equation (Equation 4) with the exact expression for Kemeny’s constant, given by Equation (Equation 3). In [17], we have already compared the exact value of K(P) with the upper bound for some real-world networks. However, the considered networks were of rather moderate size (N≤754). Here, we will assess the performance of $K_{U}\left(P\right)$ on real-world networks of sizes up to around 365 K nodes and 1.72 M edges.

Finally, in addition to Equation (Equation 4), we also assess the performance of an upper bound suggested by de Vriendt [19] based on the so-called resistance radius of a graph:(5)$$K\left(P\right)\leq L\sigma^{2}\equiv K^{*},$$
where the resistance radius $\sigma^{2}$ is defined as(6)$$\sigma^{2}=\frac{1}{2}{\left(u^{T}\Omega^{-1}u\right)}^{-1},$$
with Ω denoting the resistance matrix and *u* the all-one vector. The upper-bound Equation (Equation 5) is tight for vertex-transitive graphs. Here, we remark that vertex-transitive graphs are rather exceptional and are typically highly symmetric; examples of vertex-transitive graphs are Cayley graphs and the Petersen graph [20]. We will show in this paper that the bound $K^{*}$ is not a good estimate for the Kemeny constant for the classes of graphs that are considered in this paper and that $K_{U}$ is in general a much better estimate.

## 2. A Family of Biregular Graphs with Diameter 2

### 2.1. Construction

The aim is to construct a family of graphs that contains the complete bipartite and (generalised) windmill graphs as special cases and is commonly known as the combination of two regular graphs, denoted $G_{1}\vee G_{2}$. We start the construction by considering a $d_{1}$-regular graph $G_{1}$ on $N_{1}$ nodes, and a $k_{2}$-regular graph $G_{2}$ on $N_{2}$ nodes. We assume $k_{1}\geq 0$ and also $k_{2}\geq 0$. Finally, we connect every node in $G_{1}$ to every node in $G_{2}$ to obtain the graph *G*. The nodes in *G* that are also in $G_{1}$ have degree $k_{1}+N_{2}$, while the nodes in $G_{2}$ have degree $k_{2}+N_{1}$. This construction yields a graph $G=G_{1}\vee G_{2}$ that is a so-called biregular graph in which all nodes of $G_{1}$ have the same degree and the same holds for all nodes of $G_{2}$; see also [21]. Only if $k_{1}+N_{2}=k_{2}+N_{1}$ is the graph *G* regular. By construction, *G* has diameter 2.

The choice of $k_{1}=0$ and $k_{2}=0$ leads to the complete bipartite graph $K_{N_{1},N_{2}}$. If we take η isolated copies of the complete graph $K_{k}$ as $G_{1}$ and an isolated node for $G_{2}$, then *G* is the windmill graph W(η,k). If instead, we let $G_{2}$ be a complete graph $K_{l}$, then *G* is a generalised windmill graph of Type I, $W^{\prime }\left(\eta ,k,l\right)$, whereas if we let $G_{2}$ consist of *l* isolated nodes, *G* is a generalised windmill graph of Type II, $W^{\prime \prime }\left(\eta ,k,l\right)$.

Figure 2 shows an example of a graph that belongs to the suggested family of graphs. Here, $G_{1}$, on the left side of the figure, is a random regular graph with $k_{1}=3$, on $N_{1}=10$ nodes, while $G_{2}$ is a graph on $N_{2}=8$ nodes, where each node has degree $k_{2}=5$. For the graph *G*, the nodes in $G_{1}$ have degree 11, while the nodes in $G_{2}$ have degree 15.

### 2.2. Tightness of the Upper Bound $K_{U}\left(G\right)$

We will now show for the family of graphs proposed in the previous subsection that the upper-bound Equation (Equation 4) for Kemeny’s constant is tight.

**Theorem** **1.**
*Consider two graphs $G_{1}$ and $G_{2}$ with all vertices in $G_{1}$ with degree $d_{1}$ and those in $G_{2}$ degree $d_{2}$. If we connect each of the vertices in $G_{2}$ with all nodes of $G_{1}$, then Kemeny’s constant K(P) for the resulting graph G is given by $K\left(P\right)=\zeta^{T}d-\frac{H(G)}{D\mu_{1}}$, that is, the upper-bound Equation (Equation 4) is tight.*


**Proof.** First, we give expressions for the average degree *D* and the heterogeneity index *H*, which appear in the upper-bound Equation (Equation 4). Denoting the degrees of the nodes in *G* in $G_{1}$ and $G_{2}$ as $D_{1}$ and $D_{2}$, respectively, we obtain(7)$$D_{1}=D\left(G_{1}\right)+N_{2}=d_{1}+N_{2}$$
and(8)$$D_{2}=D\left(G_{2}\right)+N_{1}=d_{2}+N_{1}$$The average degree of *G*, D(G), which we abbreviate for notational convenience to *D*, is defined by(9)$$D=\frac{D_{1}N_{1}+D_{2}N_{2}}{N_{1}+N_{2}}.$$The heterogeneity index H(G), a metric which quantifies the variability of the degree distribution (see [16]), is defined as follows:(10)$$H\left(G\right)=\frac{1}{N}\sum_{i=1}^{N}{\left(D_{i}-D\right)}^{2},$$
where $D_{i}$ denotes the degree of node *i* in graph *G*. Using the expressions for degrees $D_{1}$ and $D_{2}$ found in (Equation 7) and (Equation 8) and expression (Equation 9) for *D*, we obtain(11)$$\begin{array}{cc} H(G) & =\frac{1}{N_{1}+N_{2}}\left(\sum_{i=1}^{N_{1}}{\left(D_{1}-D\right)}^{2}+\sum_{i=N_{1}+1}^{N_{2}}{\left(D_{2}-D\right)}^{2}\right)\\ \; & =\frac{1}{N_{1}+N_{2}}\left(N_{1}{\left(D_{1}-D\right)}^{2}+N_{2}{\left(D_{2}-D\right)}^{2}\right)=\frac{N_{1}N_{2}{\left(D_{1}-D_{2}\right)}^{2}}{{\left(N_{1}+N_{2}\right)}^{2}}. \end{array}$$We will now prove the statement by first calculating the Laplacian matrix *Q* for the graph *G*, which has the following special structure:(12)$$\begin{array}{c} Q=\left(\begin{array}{cc} A_{1} & -J_{N_{1}\times N_{2}}\\ -J_{N_{2}\times N_{1}} & A_{2} \end{array}\right), \end{array}$$
where $J_{N_{2}\times N_{1}}$ is an all-one $N_{2}\times N_{1}$ matrix, and the square matrices $A_{1}$ and $A_{2}$ are defined as(13)$$\begin{array}{c} A_{1}=Q_{G_{1}}+N_{2}I_{\left[N_{1},N_{1}\right]}\\ A_{2}=Q_{G_{2}}+N_{1}I_{\left[N_{2},N_{2}\right]}, \end{array}$$
where $Q_{G_{1}\left(G_{2}\right)}$, is the Laplacian of graph $G_{1}$ ($G_{2}$), and $I_{\left[N_{1},N_{1}]\right]},I_{\left[N_{2},N_{2}\right]}$ denote the identity matrices of size $N_{1}\times N_{1}$ and $N_{2}\times N_{2}$, respectively. The decomposition of *Q* into 4 blocks can be understood by realising that the upper right-hand block, $-J_{N_{1}\times N_{2}}$, represents the $N_{2}$ links that exist between each vertex of $G_{1}$ and all the vertices of $G_{2}$. Since *Q* is a Laplacian matrix, we have to ensure that all rows sum up to zero, which can be achieved by adding $N_{2}$ to each of the diagonal entries of the $N_{1}\times N_{1}$ block in the upper left-hand corner, that is, the block $A_{1}$ should be as defined above. Analogously, we find that the lower left-hand and right-hand blocks should be equal to $-J_{N_{2}\times N_{1}}$ and $A_{2}$, respectively.Two eigenvectors, $v_{1}$ and $v_{N}$, can be found by inspection. $v_{N}={\left(1,\cdots ,1\right)}^{T}$, which corresponds to eigenvalue $\mu_{N}=0$, and  $v_{1}={\left(N_{2},\cdots ,N_{2},-N_{1},\cdots ,-N_{1}\right)}^{T}$, which has $N_{1}$ entries equal to $N_{2}$ and $N_{2}$ entries equal to $-N_{1}$ and corresponds to $\mu_{1}=N_{1}+N_{2}$.Because the largest Laplacian eigenvalue is upper-bounded by *N*, the number of nodes in a graph (see [22]), we directly obtain that $\mu_{1}$ is the largest eigenvalue of *Q*. Combining this with Equations (Equation 9)–(Equation 11), we obtain(14)$$\frac{H(G)}{D\mu_{1}}=\frac{N_{1}N_{2}{\left(D_{1}-D_{2}\right)}^{2}}{{\left(N_{1}+N_{2}\right)}^{2}\left(D_{1}N_{1}+D_{2}N_{2}\right)}.$$Since eigenvectors corresponding to different eigenvalues are all orthogonal and those corresponding to the same eigenvalues can be chosen to be orthogonal, due to the symmetry of *Q*, all eigenvectors $v={\left(x_{1},x_{2},\cdots ,x_{N_{1}+N_{2}}\right)}^{T}$ that are not equal to $v_{1}$ or $v_{N}$ are subject to$$\begin{array}{cc} x_{1}+x_{2}+\cdots +x_{N_{1}+N_{2}} & =0\\ N_{2}\left(x_{1}+\cdots +x_{N_{1}}\right)-N_{1}\left(x_{N_{1}+1}+\cdots +x_{N_{1}+N_{2}}\right) & =0, \end{array}$$
which leads to(15)$$\begin{array}{cc} x_{1}+x_{2}+\cdots +x_{N_{1}} & =0\\ x_{N_{1}+1}+\cdots +x_{N_{1}+N_{2}} & =0. \end{array}$$We next turn to the expression $d^{T}Q^{†}d$, where $Q^{†}=\sum_{i=1}^{N_{1}+N_{2}-1}\frac{1}{\mu_{i}}{\hat{v}}_{i}{\hat{v}}_{i}^{T}$ where $\mu_{i}$ is the *i*-th eigenvalue of *Q* and ${\hat{v}}_{i}$ is the normalised eigenvector. The conditions for the eigenvectors (Equation 15) imply that all terms in the expression $d^{T}Q^{†}d$ vanish except the term associated with $v_{1}$. More precisely, we find that(16)$$\begin{array}{cc} d^{T}Q^{†}d=\sum_{i=2}^{N_{1}+N_{2}-1}\frac{{\left(d^{T}{\hat{v}}_{i}\right)}^{2}}{\mu_{i}}+\frac{{\left({\hat{v}}_{1}^{T}d\right)}^{2}}{\mu_{1}} & =\frac{N_{1}N_{2}{\left(D_{1}-D_{2}\right)}^{2}}{{\left(N_{1}+N_{2}\right)}^{2}}, \end{array}$$
where $d={\left(D_{1},D_{2}\right)}^{T}$, so the first $N_{1}$ components all have degree $D_{1}$ and the remaining components have degree $D_{2}$, which implies $d^{T}v_{i}=0$ by Equation (Equation 15). Finally, because from $D=\frac{2L}{N}$, we obtain(17)$$2L=D_{1}N_{1}+D_{2}N_{2}$$
it follows that $\frac{d^{T}Q^{†}d}{2L}$ equals Equation (Equation 14), which proves the proposition. □

### 2.3. Some Examples

As a first example, we consider the graph depicted in Figure 2, where $N_{1}=10$, $d_{1}=3$, $N_{2}=8$ and $d_{2}=5$. Using Python (https://www.python.org/) code, we have evaluated both *K* and $K_{U}$. For this network, we obtain K=16.33864, which is equal to $K_{U}$ to numerical precision, as should be according to Theorem 1. On the other hand, the upper bound $K^{*}$ based upon the resistance radius gives $K^{*}=19.29615$, which is reasonably close to the actual value.

Next, we consider a graph where $N_{1}=50$, $d_{1}=4$, $N_{2}=10$ and $d_{2}=6$; see Figure 3. Here, we get K=59.19805, and again, *K* and $K_{U}$ are numerically extremely close. On the other hand, for this graph, the bound Equation (Equation 5) is two orders larger than the actual value: $K^{*}=533.03153$.

As a final example, we consider the case where $N_{1}=100$, $d_{1}=10$, $N_{2}=20$ and $d_{2}=8$; see Figure 4. Now, K=119.24078 and again *K* and $K_{U}$ are equal to numerical precision. Again, the bound based on the resistance radius is much higher: $K^{*}=1652.63986$.

We end this subsection by noting that the choice for the examples in this subsection was rather arbitrary. We also ran our Python script on several other graphs with sizes up to 1500 nodes. Each time, it yielded the same result: *K* and $K_{U}$ have values that are numerically very close (see also [4,17] for more numerical comparisons), while the upper bound $K^{*}$ exceeds Kemeny’s constant by a few orders.

## 3. Graphs with Diameter 2 for Which $K_{U}\left(P\right)$ Is Not Tight

### 3.1. Bimodal Graphs with Diameter 2 for Which Equation (4) Is Not Tight

The numerical results of the examples on biregular graphs with diameter 2 from the previous section showed that in all these cases, the approximation of *K* by $K_{U}$ is actually exact. In other words, the bound $K_{U}$ is tight in these cases. Therefore, one might be tempted to believe that Equation (Equation 4) is tight for all biregular graphs with diameter 2. In this section, we prove that this is not the case by giving some counterexamples.

The simplest counterexample we could find consists of the cycle graph $C_{5}$ with an additional link; see Figure 5.

For this graph, we get K=3.71212, $K_{U}=3.72380$ and $K^{*}=4.21488$. There is a simple procedure to check whether or not a biregular graph *G* with diameter 2 belongs to the graph family constructed in the previous section. First, partition the nodes into two sets $S_{1}$ and $S_{2}$ where all the nodes in the set $S_{1}$ have degree $D_{1}$, while all the nodes in the set $S_{2}$ have degree $D_{2}$. Next, verify if the number of links between the 2 sets is $|S_{1}\times S_{2}|$ and all nodes of $S_{1}$ are linked to all nodes of $S_{2}$. If this is not the case, the graph $G\neq S_{1}\vee S_{2}$. In the other case, remove all $|S_{1}\times S_{2}|$ links between the sets $S_{1}$ and $S_{2}$. If the remaining two graphs are not both regular, then the original graph *G* does not belong to the family constructed in the previous section, that is, $G\neq S_{1}\vee S_{2}.$

The second counterexample is constructed by adding a link to the Petersen graph; see Figure 6.

For this graph, we get K=9.76597, $K_{U}=9.77389$ and $K^{*}=10.26963$.

### 3.2. Non-Biregular Graphs with Diameter 2

We now give an example of a non-biregular graph with diameter 2, for which the upper-bound Equation (Equation 4) also does not equal Kemeny’s constant. We construct the graph by first taking a complete graph $K_{N}$ on *N* nodes. Next, we add one node and connect it to one node in $K_{N}$ and therefore the resulting graph has diameter 2. The resulting graph has N−1 nodes with degree N−1, one node with degree *N*, and one node with degree 1. Figure 7 shows an example with N=10.

Applying Equation (Equation 3), we get K=9.26522, while the upper bound of Equation (Equation 4) gives $K_{U}=9.83439$, while $K^{*}=31.28000$.

## 4. Regular Graphs

In this section, we consider regular graphs on *N* nodes with degree *r*. In this case, the relation between Kemeny’s constant and the effective graph resistance was shown [23] to be(18)$$K\left(P\right)=\frac{r}{N}R_{G},$$
where $R_{G}$ denotes the effective graph resistance. Next, we show that for these graphs, the upper-bound Equation (Equation 4) is also tight. For this, we will use the following expression for the effective graph resistance (see [4]):(19)$$R_{G}=N\sum_{i=1}^{N}Q_{ii}^{†}.$$

For *r*-regular graphs, H=0, and therefore Equation (Equation 4) gives(20)$$K_{U}=\zeta^{T}d=r\zeta^{T}u=r\sum_{i=1}^{N}Q_{ii}^{†}=\frac{r}{N}R_{G},$$
hence $K_{U}=K$ according to Equation (Equation 18).

As an example, we consider a random 3-regular graph on 100 nodes (see Figure 8), which has a diameter 10. We get numerically K=195.30524, which is indeed equal to $K_{U}$ up to the numerical precision of $10^{-17}$. Applying Equation (Equation 5) gives $K^{*}=218.32805$. In this case, the upper-bound Equation (Equation 5) is not tight because the graph is not vertex-transitive.

## 5. Complexity for the Computation of $K_{U}\left(P\right)$

The time complexity of K(P), computed via Equation (Equation 3), is dominated by the Laplacian pseudo-inverse, which is as expensive as performing a dense matrix multiplication and takes $O(N^{3})$ in practice with standard tools. On the other hand, the time complexity of $K_{U}\left(P\right)$ mainly depends on two operations: computing the largest Laplacian eigenvalue and performing the dot product of a degree vector and the diagonal element vector of the Laplacian pseudo-inverse. Interestingly, to compute $K_{U}\left(P\right)$, we can avoid the full pseudo-inversion as it only requires the diagonal elements of the Laplacian pseudo-inverse. Algorithms that approximate the diagonal (or the trace) of matrices often use iterative methods, sparse direct methods [24], Monte Carlo [25] or deterministic probing techniques [26]. Although faster than computing the full inversion, these approaches are still time-consuming in practice for large graphs [27]. For that reason, we employ a recently proposed algorithm that approximates the diagonal entries of the Laplacian pseudo-inverse using combinatorial connections [27]. This algorithm exploits the relation between effective resistance and the pseudo-inverse Laplacian. In order to calculate the diagonal elements of $Q^{†}$, it is sufficient to compute the electrical farness $f_{\mathrm{el}}\left(u\right)$ of each node *u* in the set of all nodes *V*; the farness is defined by$$f_{\mathrm{el}}\left(u\right)=\sum_{v\in V/\{u\}}R\left(u,v\right)=NQ_{uu}^{†}+\mathrm{Tr}\left(Q^{†}\right)$$Here, R(u,v) is the effective resistance between node *u* and *v*, which is the potential difference between *u* and *v* when a unit current is injected in graph *G* at node *u* and extracted at node *v* [28]. Rather than calculate R(u,v) for each pair of nodes, we sample a set of uniform spanning trees. This approach provides a probabilistic absolute approximation guarantee.

The algorithm’s time complexity is summarised in the following proposition:

**Proposition** **1**([27]). *Let G=(V,E) be an undirected and weighted graph with N nodes and L edges. The sampling algorithm, briefly described above, gives an approximation of the diagonal elements of $Q^{†}$ with absolute error ±ϵ with probability 1−δ in an expected time $O(L\cdot {\mathrm{ecc}}^{3}\left(u\right)\cdot \epsilon^{-2}\cdot \log\left(L/\delta \right))$, where ecc(u) is the length of the longest shortest path (eccentricity) starting in a selected node u. For small-world graphs and δ=1/N (for high probability), this yields a time complexity of $O(L\log^{4}N\cdot \epsilon^{-2})$.*

For networks that have small-world characteristics, a common feature for many real-world networks [29], the above algorithm obtains a ±ϵ-approximation with high probability, in a time that is linear in *L* up to polylogarithmic terms and quadratic in 1/ϵ. Furthermore, computing the largest Laplacian eigenvalue does not change the overall complexity bound. More precisely, this step often takes O(L) time for sparse matrices using standard iterative methods, such as the Lanczos algorithm [30]. In general, the actual running time for this step highly depends on the desired accuracy and the eigenvalue distribution of the involved matrix. Overall, the complexity bound for computing $K_{U}\left(P\right)$ for small-world graphs using the above techniques is linear in the number of links *L* (up to a polylogarithmic factor).

## 6. Analysis of Some Large Real-World Networks

In this section, we analyse the performance of our proposed bound, $K_{U}\left(P\right)$, compared to Kemeny’s constant, K(P), in terms of accuracy and running time results. For $K_{U}\left(P\right)$, our implementation uses the NetworKit [31] graph library to compute the diagonal elements of $Q^{†}$ (via the algorithm of Angriman et al. [27]) and the Slepc library (https://slepc.upv.es/) (accessed on 2 December 2024) to compute the largest Laplacian eigenvalue. K(P), in turn, is computed via Equation (Equation 3) and our implementation uses the Eigen library (http://eigen.tuxfamily.org) (accessed on 2 December 2024) to compute the entire pseudo-inverse, $Q^{†}$. We do not include any comparisons against $K^{*}$ since, computationally, it is as expensive as the exact computation of Kemeny’s constant. Our test machine is a shared-memory server with a 2x 18-Core Intel Xeon 6154 CPU and a total of 1.5 TB RAM. To ensure reproducibility, experiments are managed by SimexPal [32]. In Table 1, we list the real-world graphs that are used in our experiments, downloaded from SNAP [33] and NR [34] public repositories. In this context, we consider as medium graphs those whose vertex count is <57 K. The largest graph has around 365 K nodes and 1.72 M edges.

For the medium graphs of Table 1, we are able to compare our bound $K_{U}\left(P\right)$ relatively to Kemeny’s constant K(P), and the results are illustrated in Figure 9. $K_{U}\left(P\right)$ is computed with different error bounds (ϵ) for the approximation of the diagonal elements (via the algorithm of Angriman et al. [27])—they correspond to the respective numbers next to the names in Figure 9. Regarding the accuracy, we observe that our approach for computing $K_{U}\left(P\right)$ is overall highly accurate for all values of ϵ and graphs. More precisely, on average (computed via geometric mean) over the medium-size graphs, our approach is 0.33% 0.27% 0.25% and 1.26% away from the exact Kemeny’s constant for ϵ=0.1, 0.3, 0.5 and 0.9, respectively. Meanwhile, the running time is on average 2, 18, 48 and 141× faster than the exact computation for each ϵ, respectively. Figure 9a shows that on individual graphs, a larger ϵ value (ϵ=0.9) may result in a slightly less accurate bound—up to 10% away from the exact value (arx). Moreover, in Figure 9b, we observe that for the inf graph, computing the exact Kemeny’s constant is much faster than computing $K_{U}\left(P\right)$ via Algorithm [27]. The primary reason for that is the small size (6K edges) for which an exact computation of the entire pseudo-inverse is still fast enough. A second reason for the slow performance of the algorithm of Angriman et al. could be due to the high diameter of the graph in question (≫logN).

In Table 2, we illustrate our results for the largest graphs of Table 1. For this experiment, we set ϵ=0.5 for the approximation of the diagonal elements of $Q^{†}$ as this offers the best trade-off between accuracy and speed, according to the previous experiment. Unfortunately, we were not able to compute exact values for Kemeny’s constant for these graphs, as all involved runs timed out at 18,000 s. This is due to the prohibitive time and space complexity of the pseudo-inversion operation required by K(P).

## 7. Conclusions

We have investigated Kemeny’s constant K(P) for a number of networks using the exact expression from [4] and compared this expression with two upper bounds: one $K^{*}\left(P\right)$ that was derived in Ref. [19] and is known to be tight for vertex-transitive graphs, and the other bound $K_{U}\left(P\right)$ was derived in [4] and is written in terms of degrees of the nodes, the diagonal elements of the pseudo-inverse Laplacian, the largest eigenvalue of the Laplacian matrix and the heterogeneity of the degrees of the nodes.

We have numerically demonstrated that the bound $K_{U}\left(P\right)$ is generally a much better approximation for K(P) than $K^{*}\left(P\right)$ for the networks that we have explored. Moreover, we have proved that for any graph *G* composed of two regular graphs $G_{1}$ and $G_{2}$ with all nodes of the graph $G_{1}$ connected to each node of $G_{2}$, the bound $K_{U}\left(P\right)$ is tight. This generalises earlier findings that the bound $K_{U}\left(P\right)$ is tight for (generalised) windmill and complete bipartite graphs.

As an illustration of the advantages of using the expression $K_{U}\left(P\right)$ to estimate the Kemeny constant, we numerically calculated the Kemeny constant for a number of real-world large networks. We find that the calculation of $K_{U}\left(P\right)$ can be performed very efficiently, displaying efficiency gains in the order of a factor 100–1000, for networks up to 57 K nodes. The upper bound can still be obtained in a reasonable time for networks up to 365 K nodes.

## Figures and Tables

**Figure 1 entropy-27-00384-f001:**
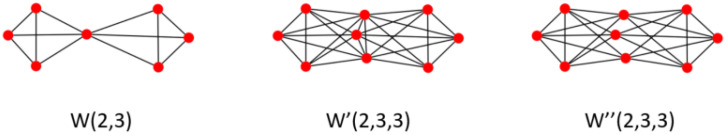
A windmill graph and generalised windmills of Types I and II.

**Figure 2 entropy-27-00384-f002:**
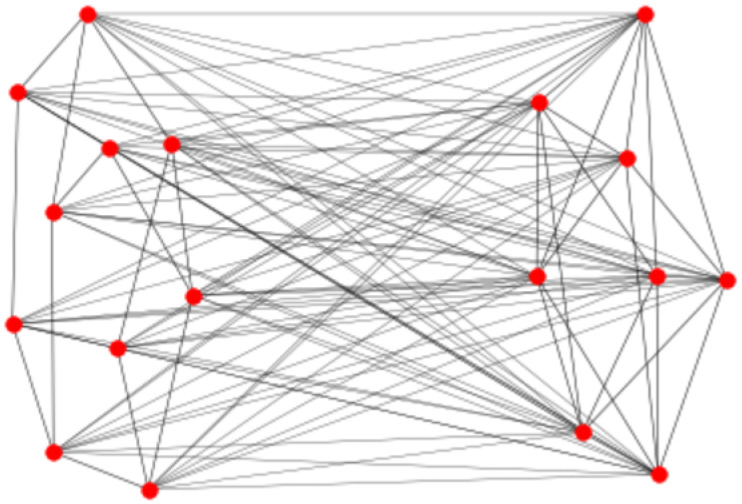
Graph *G* on 18 nodes, where $G_{1}$ is a random 3-regular graph on 10 nodes, and $G_{2}$ is a 5-regular graph on 8 nodes.

**Figure 3 entropy-27-00384-f003:**
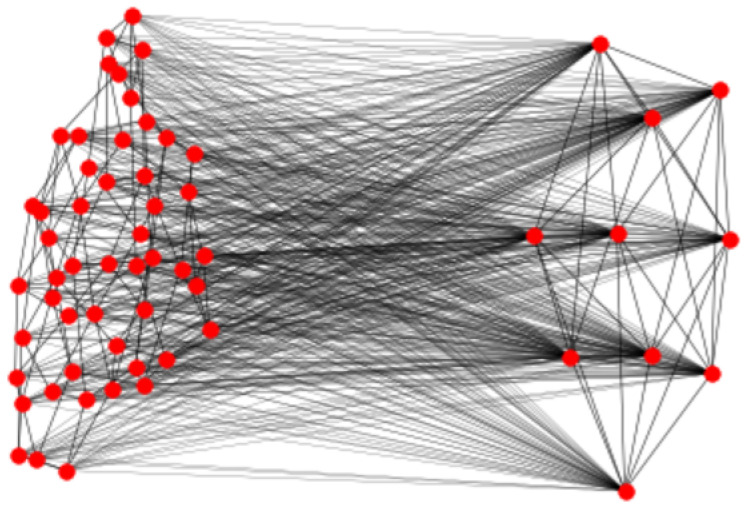
Graph *G* on 60 nodes, where $G_{1}$ is a random 4-regular graph on 50 nodes, and $G_{2}$ is a random 6-regular graph on 10 nodes.

**Figure 4 entropy-27-00384-f004:**
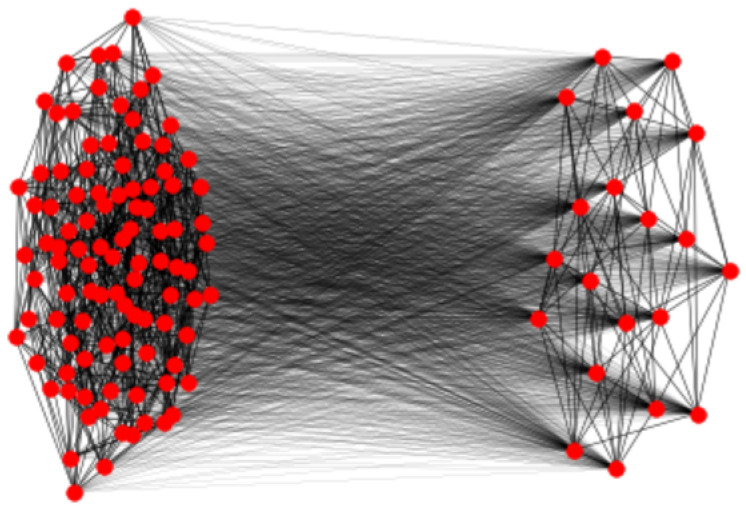
Graph *G* on 120 nodes, where $G_{1}$ is a random 10-regular graph on 100 nodes, and $G_{2}$ is a random 8-regular graph on 20 nodes.

**Figure 5 entropy-27-00384-f005:**
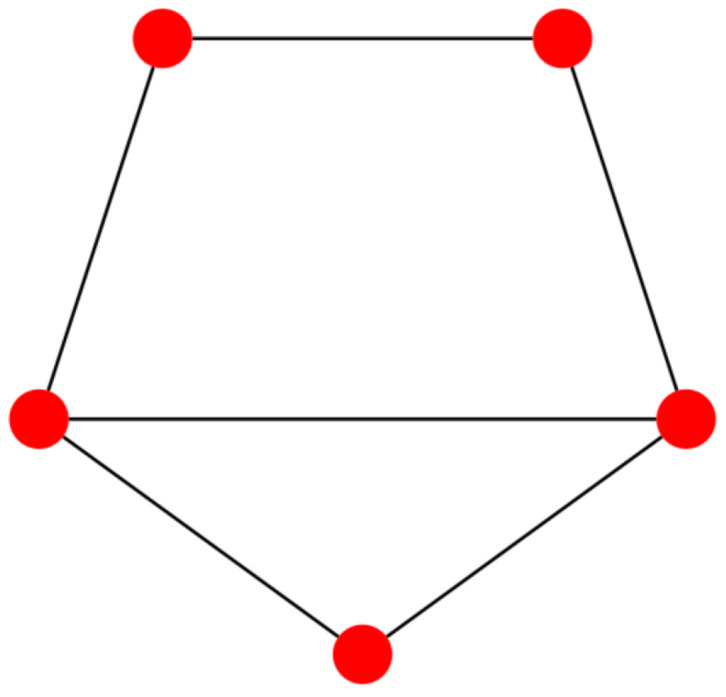
Smallest biregular graph with diameter 2 for which the upper-bound Equation (Equation 4) is not tight.

**Figure 6 entropy-27-00384-f006:**
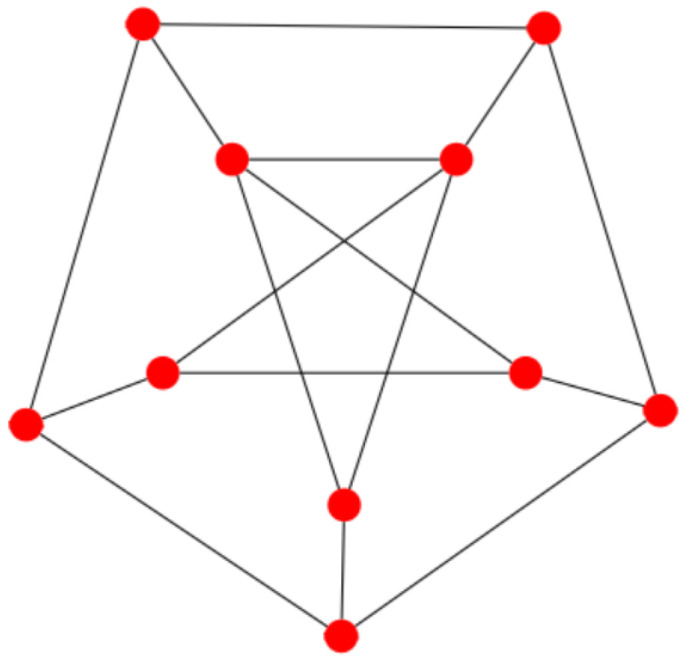
Petersen graph with one additional link.

**Figure 7 entropy-27-00384-f007:**
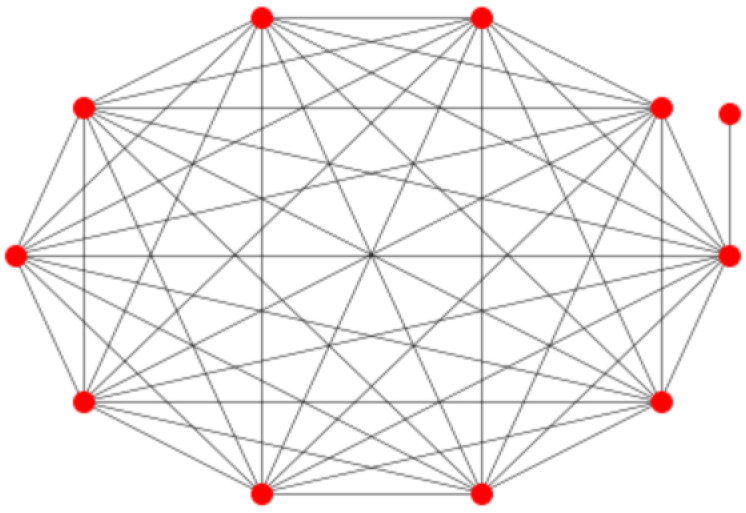
Non-biregular graph with diameter 2.

**Figure 8 entropy-27-00384-f008:**
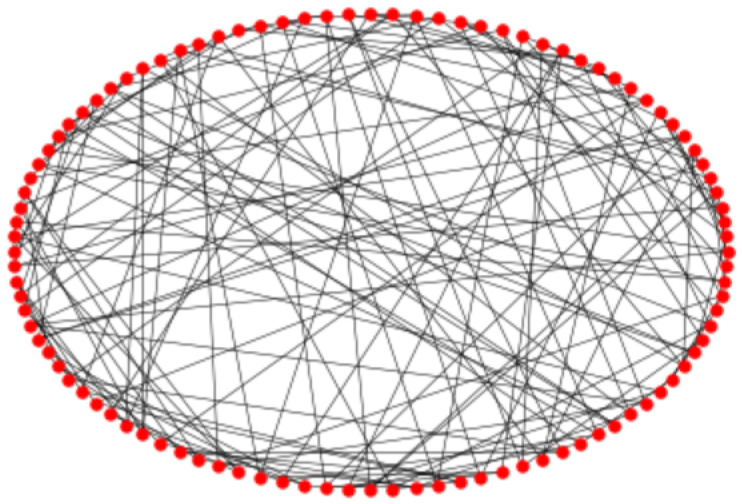
Random 3-regular graph on 100 nodes.

**Figure 9 entropy-27-00384-f009:**
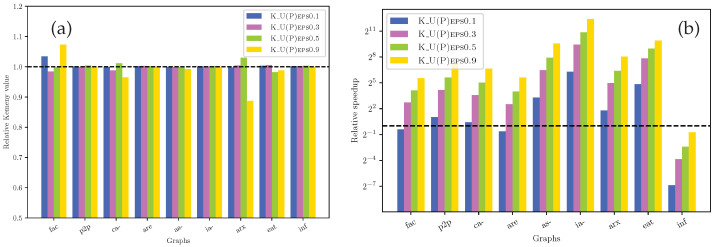
Relative quality (**a**) and speedup (**b**) results (per graph) for computing $K_{U}\left(P\right)$ for medium graphs (n<57 K) of Table 1. Results are relative to exact computation of K(P).

**Table 1 entropy-27-00384-t001:** Summary of graph instances, providing (in order) network name, type, abbreviation, vertex count, and edge count.

Graph	Type	ID	|V|	|E|
inf-power	infrastructure	inf	4 K	6 K
facebook-ego-combined	social	fac	4 K	8.8 K
p2p-Gnutella04	internet	p2p	10 K	39 K
ca-HepPh	collaboration	ca-	11 K	117 K
arxiv-astro-ph	collaboration	arx	17 K	196 K
eat	words	eat	23 K	297 K
arenas-pgp	infrastructure	are	24 K	10 K
as-caida20071105	internet	as-	26 K	53 K
ia-email-EU	communication	ia-	32 K	54.4 K
loc-brightkite	social	lob	57 K	213 K
soc-Slashdot0902	social	soc	82 K	504 K
flickr	images	fli	106 K	2.31 M
livemocha	social	liv	104 K	2.19 M
loc-gowalla-edges	social	log	196 K	950 K
web-NotreDame	web	web	325 K	1.09 M
citeseer	citation	cit	365 K	1.72 M

**Table 2 entropy-27-00384-t002:** Absolute results for $K_{U}\left(P\right)$ on the largest graphs of Table 1. Comparison to K(P) is prohibitive due to the (large) size of the graphs in question.

Graph	$K_{U}\left(P\right)$	Time (h:min:s)
lob	80,903	48.83 s
soc	96,102	50.87 s
fli	122,185	1 min:38.11 s
liv	120,525	37.07 s
log	271,577	5 min:10.77 s
web	1,009,760	1 h:11 min:19.36 s
cit	508,244	1 h:16 min:11.51 s

## Data Availability

No new data were created or analyzed in this study. Data sharing is not applicable to this article.

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
