# Peer review of "Further Exploration of an Upper Bound for Kemeny’s Constant"

_entropy, 2025, doi:10.3390/e27040384_

Round 1

Reviewer 1 Report

Comments and Suggestions for Authors

See attachement

Reviewer 2 Report

Comments and Suggestions for Authors

In itself, this paper presents a valuable contribution to the field of spectral graph theory and its applications. The authors investigate Kemeny's constant, a crucial metric in network analysis, focusing on developing and analyzing an upper bound for its computation. The work is well-structured and the methodology is clearly presented. The numerical paragraphs are impeccable. However, I have a major concern about why this is suitable for a journal called "Entropy". I have not directly seen any entropy-related topic in this paper.

The author are therefore invited to provide a clear connexion with this topic.

In addition, some aspects could be strengthened. In particular:

  • In the introduction, the (very first) definition of K(P) is not expressed. A more comprehensive discussion on what K(P) is and what it does (and why it is important) is more than welcome.
  • From lines 26 to 29, this should be referenced.
  • Replace "links" by "edges" in line 32
  • Equation (2) not not clear. \mu_1 is a function of G?
  • Define a "tight" upper bound in line 40.
  • End of Section 1. There is no discussion on which of the two upper bounds K_U or K* is best.
  • Theorem 1: careful with notations! d_1 (d_2) have been changed with k_1 (k_2).
  • I don't see why Eqs (5) and (6) are relevant?
  • Here again careful with notations. "D(G_1)", "D"... Is D a function of G or and average?
  • You do not deduce Eq(7) from Eqs (5) and (6), thus the "therefore" doesn't look to have anything to be there.
  • Eq. (10) should be proven. Generally speaking, there is no reason to remain obscur even if easy. It makes the text more fluent (and beautiful).
  • Type in line 107
  • line 113: "largest Laplacian eigenvalue of Q" incorrectly said.
  • Before line 115, "imply that all terms vanish except the term with v_1". Please make it fluent (prove it).
  • Section 2.3 : how is K-K_U converging when N_1, N_2 tend to +\infty?
  • So K* is not a good upper bound here?
  • Section 3.1: "Given the results [...] with diameter 2". Why this intuition? Please explain.
  • Line 146: typo "the the"
  • Remind the graphs for which K* is a good upper bound in the intro. You only say this in the conclusion, but throughout the text I was really wondering why you were wasting your time with K*.
  • Section 5: should be more rigorously written. In particular, I don't know why you stated Proposition 1 but you don't look to use is anywhere in the text. Also, while the authors discuss the computational complexity, a more formal analysis would strengthen the paper. They mention "nearly-linear" complexity but lack a more precise characterization.
  • Table 1: I was curious about the bitcoin network?
  • Table before the conclusion: show time in format "hour:min:sec"

Providing an entropy connection and the resolution of the above problems would lead me to accept the paper.

Round 2

Reviewer 2 Report

Comments and Suggestions for Authors

The authors have reviewed the suggestions overall from the first suggestions. The paper is close to its final version, and is clear and good! I suggest the minor following points.

  • in Equation (4), write \mu_1(G)
  • I still don't understand why the authors deduce Eq.(7) from Eqs.(5-6). To me the latter equations are not needed as Eq.(7) is simply and average. The "therefore" looks inappropriate.
  • I don't see any additional comment on the convergence of K and K as asked in my previous review. This is not regarding Prop 1 about convergence of the algorithm, but close are K and K with n.
